# Certificateless broadcast signcryption scheme supporting equality test in smart grid

**Shufen Niu, Runyuan Dong**  **\*, Lizhi Fang**

College of Computer Science and Engineering, Northwest Normal University, Lanzhou, Gansu, China

\* 1981990687@qq.com

## Abstract

With the development of cloud computing and the application of Internet of Things (IoT) in the smart grid, a massive amount of sensitive data is produced by the terminal equipment. This vast amount of data is subject to various attacks during transmission, from which users must be protected. However, most of the existing schemes require a large amount of network bandwidth resources and cannot ensure the receiver's anonymity. To solve these shortcomings, we construct a broadcast signcryption scheme supporting equality test based on certificateless cryptosystem. The scheme employs a symmetric encryption algorithm to improve encryption and transmission efficiency; The Lagrange interpolation theorem is used to encrypt the user's identity to ensure the privacy preservation of terminal devices; And a trusted third party is used to eliminate duplicated ciphertext for identical messages using an equality test, resulting in efficient network bandwidth utilization. Experimental analysis shows that our work has greater advantages in the field of practical broadcast services.

## Introduction

The use of intelligent Internet of Things (IoT) devices brings great convenience to data communication. Wireless sensor network has been widely applied in the smart grid. Considering wireless sensor networks transmit information over public channels, and that power data and sensitive information may be distorted in the transmission process, there are security risks associated with the smart grid. Many fields of study use it in conjunction with cryptography algorithms to guarantee the security of sensitive information. Amos Fiat and Moni Naor [1] first proposed broadcast encryption, which is suitable for one-to-many communication where the broadcaster transmits the encrypted data to an authorized receiver. Each receiver can obtain ciphertext and decrypt messages with their private key. Compared with the traditional one-to-one encryption model, broadcast encryption can reduce the computation and communication overhead. Therefore, it has significant application and value in IoT scenarios [2–4]. Unfortunately, they are unable to guarantee the privacy of users or the transmission of sensitive information. As the information collected by IoT equipment is sensitive, users expect a guarantee of the security of data transmission and communication. A certificateless cryptosystem is proposed to deal with the key escrow problem. It can realize efficient and secure transmission of broadcast ciphertext [5–8]. Considering the same broadcast ciphertext may be

**Data Availability Statement:** All relevant data are within the experimental analysis section of the manuscript.

**Funding:** This work was supported by the National Natural Science Foundation of China

(No.62241207) and Gansu Science and Technology Program (22JR5RA158). The funders had no role in study design, data collection and analysis, decision to publish, or preparation of the manuscript.

**Competing interests:** The authors have declared that no competing interests exist.

generated by different encryption methods, it will occupy bandwidth resources on resource-limited devices, limit the applicability of the application environment, and cause great inconvenience and waste of space. The ciphertext equality test [9–11] can match ciphertexts on broadcasters and cloud servers, so as to realize the de-duplication of redundant copies and save bandwidth resources.

Currently, there exist malicious attackers in the smart grid, causing the smart grid to face some security threats, such as user forging smart meter data, unauthorized user access to sensitive information leading to privacy leakage, and malicious attacker stealing data when wireless sensor networks transmit information over public channels. Thus, the scheme demands the intelligent power supply system encrypt data and send ciphertext to users in the form of broadcasting, so as to transmit users' power information efficiently and securely. Broadcast signcryption achieves data sharing between broadcast servers and authorized receivers. Unfortunately, many existing broadcast signcryption schemes have some shortcomings. They are unable to realize the private preservation of receiver identities and need a lot of bandwidth resources.

Based on the current security threat in smart grid and the shortcomings of existing schemes, we propose a broadcast signcryption scheme that supports equality test based on certificateless cryptosystem. First, the scheme solves the key escrow problem in identity-based cryptosystem by using certificateless cryptosystem and ensures the receiver's anonymity by using the Lagrange interpolation theorem. Second, our proposed scheme can be proven secure under the Random Oracle Model. In addition, the proposed scheme realizes the function of data de-duplication by using equality test and lightweight broadcast signcryption by reducing computation cost.

## Motivation and contributions

To realize the private preservation of smart meter's identities and the confidentiality of sensitive information, while also saving bandwidth resources, we propose a broadcast signcryption scheme supporting equality test based on certificateless cryptosystem. The main contributions of our work as follows:

- The scheme ensures the privacy of the user's identity. Not only are illegal receivers unable to obtain the sender's identity but the receiver also do not know the other receiver's identity.

- The proposed scheme uses equality test to realize the function of data de-duplication. To achieve efficient utilization of network bandwidth, the duplicate ciphertext of the information generated by different encryption methods is de-duplicated by a trusted third party.

- We realize lightweight broadcast signcryption by reducing the bilinear pairing operation with a high computation cost in unsigncryption. The experimental analysis showed that the computing efficiency was higher than existing schemes, and had greater advantages in practical applications.

## Organization

The organization of this paper as follows. We survey the related works in Section 2. In Section 3, we briefly describe the background. Our scheme and correctness are present in Section 4. Security proof is given in Section 5. In Section 6, we present the performance evaluation. Finally, we conclude the work in Section 7.

## Related works

Duan et al. [12] first construct the broadcast signcryption scheme in combination with the signcryption algorithm and the broadcast transmission. Unfortunately, the scheme does not

meet the security requirements of adaptive chosen-ciphertext attack. Based on the problem of one-to-one single transmission in the traditional signcryption scheme, broadcast signcryption solves the shortcomings of communication efficiency in information transmission. Since the invention of broadcast signcryption, many academics and practitioners have propose the scheme to meet various security performance requirements [13, 14]. Zhang et al. [13] construct a signcryption scheme that resists quantum attacks based on lattice and identity cryptosystem. [14, 15] designed the efficient signcryption algorithm that allowed the sender to transmit multi-messages to multi-receivers and analyzed the efficiency of each scheme. Qiu et al. [15] design a broadcast scheme based on certificateless cryptosystem and applied it to the IoT, lowering the computation cost of the receiver by outsourcing the gateway signature verification operation. However, there is the problem of key escrow. Peng et al. [16] connected the edge node with the IoT device. Edge computing can reduce the computation burden of terminal devices and the delay of network transmission. However, the ciphertext of this scheme hasn't the authorization set of the receiver. Due to the risk of location privacy leakage in the charging process of electric vehicles. Kumar et al. [17] design an electric vehicle charging framework combined with grid encryption technology. Alagarsamy et al. [18] propose an Exponentiated Multilinear Vectorized Certificateless Signcryption (EMV-CLSC) scheme, which reduces memory usage when processing multiple data and improves computation efficiency. [19–21] propose lightweight and efficient access control signcryption schemes based on the certificateless cryptosystem. Ullah et al. [20] propose an anonymous certificateless signcryption scheme using elliptic curves to guarantee security requirements in Internet of Vehicles, but this scheme only signcrypt single message and is not suitable for the multi-message environments. Sarvesh et al. [22] present a multi-signcryption scheme with public verifiability to reduce the threats of private key escrow and replay attacks. Unfortunately, These schemes fail to consider the processing of redundant data generated by different encryption methods for the same information. Luo et al. [23] propose the signcryption scheme for data communication between different network domains, but can't ensure the privacy of receivers. Khan et al. [24] set a smaller key unit based-identity signcryption, which is not applicable to equipment with limited resources, and there is the risk of the receiver's privacy leaking. Mandal et al. [25] design a user access control scheme that fails to achieve the receiver's privacy preservation. Shen et al. [26] propose a lightweight and secure data transmission protocol for wireless body area networks, which support the multidisciplinary treatment but exist a risk of leakage of the partial private key.

Aiming at addressing the shortcomings and improving the efficiency of existing schemes, we propose a broadcast signcryption scheme that supports equality test based on certificateless cryptosystem. Our proposed work ensures the receiver's anonymity and information integrity and confidentiality, while also the proposed scheme realizes the function of data de-duplication by using equality test and lightweight broadcast signcryption by reducing computation cost.

## Background

### Hard problems

We give several hard problems to demonstrate the security of our work.

**Decision Diffie-Hellman (DDH) problem** [9]. Given $P, aP, bP, W \in G$, where $a, b \in Z_p^*$, it is hard for the probabilistic polynomial time (PPT) to determines whether $W = abP$ with non-negligible advantage.

**Bilinear Diffie-Hellman (BDH) problem** [22]. Given $P, aP, bP, cP \in G$, where $a, b, c \in Z_p^*$ and $P$ denotes the generator of group $G$, compute $e(P, P)^{abc} \in G_T$, if $Pr[A(P, aP, bP, cP) = e(P, P)^{abc}] \geq \varepsilon$, the advantage of the adversary to solve the BDH problem.

**Computational Diffie-Hellman (CDH) problem** [23]. Given $aP, bP \in G$, compute $abP$ element where $a, b \in Z_p^*$ are unknown and $P$ denotes the generator of group $G$.

**Decision Bilinear Diffie-Hellman (DBDH) problem**. For $\forall a, b, c \in Z_p^*$, $\exists P, aP, bP, cP \in G$, it is hard for PPT to distinguish $e(P, P)^{abc}$ by a non-negligible advantage.

## System model

The smart power grid relies on intelligent technology, such as wireless sensors, to realize information collection and data transmission. As can be seen from Fig 1, detailed information is collected through sensor equipment. It is assumed that the application field deploys wireless sensor network nodes in multiple power jurisdictions and monitoring areas. The wireless sensor network nodes collect power data and other data in real-time and upload them to the aggregation base station. Considering wireless sensor networks transmit information over public channels, and that power data and sensitive information may be distorted in the transmission process, there are security risks associated with the smart grid. Many fields of study use it in conjunction with cryptography algorithms to guarantee the security of sensitive information.

As is shown in Fig 2, our proposed system model includes five entities: Intelligent power supply system, Trusted Third Party (TTP), Key Generation Center (KGC), Smart meter and Cloud Server (CS). After the KGC generates public and private key for the intelligent power supply system and the smart meter, the intelligent power supply system will signcrypt the collected information such as the meter operation status and smart meter data, and then sends ciphertext to CS and TTP for ciphertext equality test. TTP deletes the duplicate ciphertext of the information generated by different encryption methods. When the CS broadcasts ciphertext to smart meter, the authorized device can unsigncrypt ciphertext using private key independently.

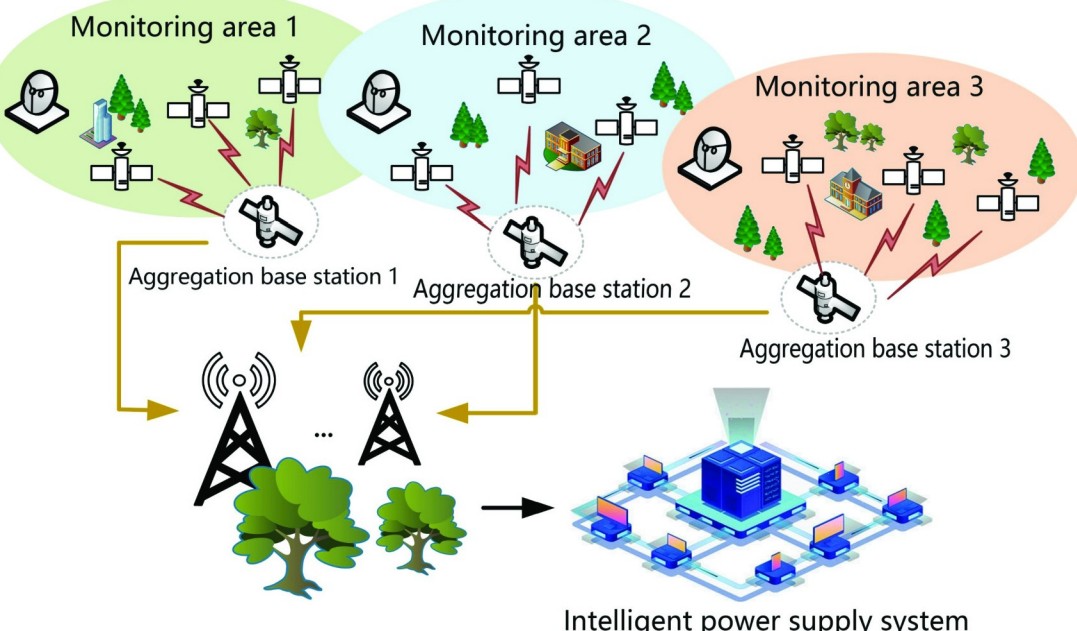

**Fig 1. Wireless sensor network transmission structure.**

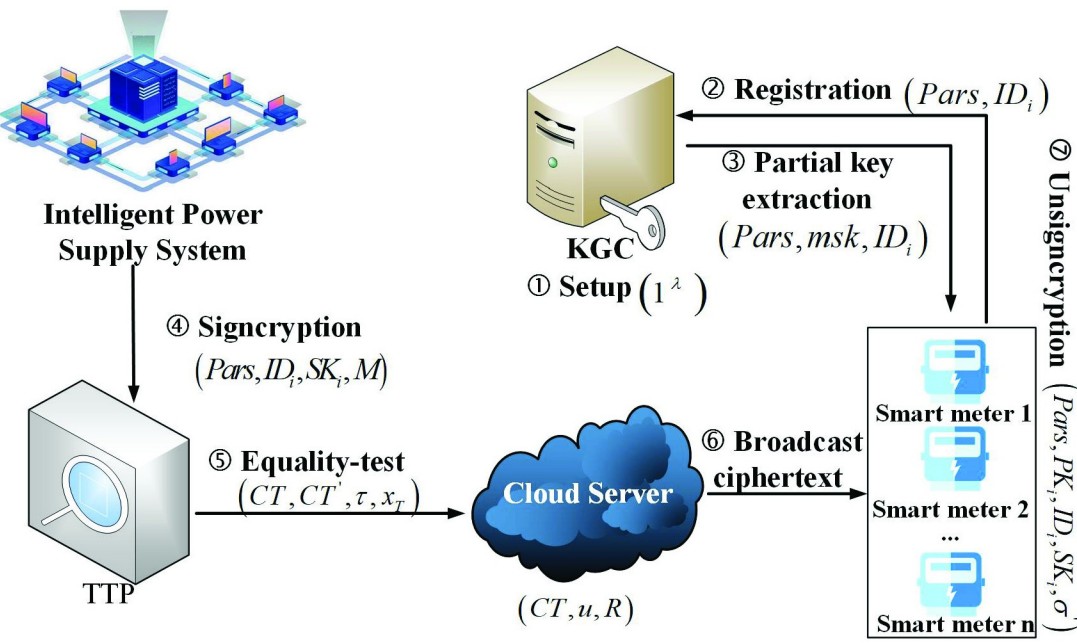

**Fig 2. System model.**

**KGC**. Assuming KGC is a fully trusted entity, the device set and the intelligent power supply system send a registration request before the broadcaster broadcasts the message. After receiving the request, the KGC generates public and private key for the smart meter and intelligent power supply system to ensure the device's legality.

**Intelligent power supply system**. After receiving the information gathered by wireless sensor network monitoring equipment, it selects a group of equipment to collect messages, encrypt message and upload it in the monitoring area, and then sends signature to the TTP for ciphertext equality test.

**TTP**. In order to rid the ciphertext of duplicated data, the TTP checks whether the received ciphertext has a copy of the same information generated by different encryption methods on the CS.

**Smart meter**. The smart meter submits registration request for legal identities to the key generation center. When the CS broadcasts ciphertext to smart meter, the authorized device can send verification information to a trusted third party for ciphertext matching. After obtaining the correct response, the ciphertext can be decrypted independently.

**CS**. The trusted third party can be operated by the CS to match the duplicate data of the ciphertext generated and transmitted throughout the entire broadcast process. Although ciphertext is stored on the CS, it can't get any information about ciphertext from the broadcaster.

## Formal definition

For the signcryption scheme supporting equality test in smart grid, we give the detailed definition as follows for algorithm:

**Setup** $(1^\lambda)$: Inputs security parameter $\lambda$, KGC returning public parameter *Pars*, TTP's private key $x_T$ and the master key $s$.

**Set secret-value** $(ID_i)$: It takes *Pars* and the receiver's identity $ID_i$ as input and returns $x_i$ as the receiver's secret value.

**Extract partial-private key** ($s$, $X_i$, $ID_i$): The inputs are the $ID_i$ of the receiver, the master key $s$, the public key of the receiver $X_i$ and the public parameters *Pars*, and it returns the partial private key $z_i$.

**Set private and public key** ($ID_i$, $z_i$, $W_i$, *Pars*): Inputs the $ID_i$ of the receiver, the partial private key $z_i$ and *Pars*, and returns the private key of receiver ($d_i$, $y_i$) and the public key of receiver ($X_i$, $Y_i$).

**Signcryption** (*Pars*, $ID_i$, $X_i$, $M$): The inputs are the public parameters *Pars*, a set $\{ID_i, X_i\}_{i=1,2,\cdots,n}$ and the message $M$. The outputs are ciphertext $\sigma$.

**Unsigncryption** ($y_i$, $\sigma$, $ID$): The inputs are the public parameter *Pars*, a private key of the smart meter $y_i$, a set $\{ID_i\}_{i=1,2,\cdots,n}$ of the receiver's identity, ciphertext $\sigma$, outputs recovered message and verify the message using the broadcaster's public key.

**Equality-test** ($CT$, $CT'$): TTP executes this algorithm. The inputs are the public parameter *Pars*, the private key $x_T$ and two ciphertexts $CT$ and $CT'$. The output is 1 if $CT$ and $CT'$ are same message generated by different encryption methods, otherwise, returns 0.

## Security model

In order to ensure broadcasting safety, the proposed work must satisfy the security of message, define indistinguishability of chosen multiple identities and chosen ciphertext attack security (IND-CMID-CCA) by polynomial time simulating the game between adversary and challenger, ensure strong unforgeability of chosen multiple identities and ciphertext attack security (SUF-CMID-CCA) and ensure anonymity of chosen multiple identities and ciphertext attack security (ANON-CMID-CCA).

**Game 1**: IND-CMID-CCA security

This game is played between adversary $\mathcal{A}$ and challenger $\mathcal{C}$ under the IND-CMID-CCA security model. The security model is defined as follows:

**Setup**: $\mathcal{C}$ takes the security parameter $\lambda$ as input and returns the public parameters *Pars* and the master key $s$, $\mathcal{C}$ sends the public parameter *Pars* to $\mathcal{A}$ and keeps $s$. Then, $\mathcal{A}$ selects a random identity from set $\{ID_i\}_{i=1,2,\cdots,n}$.

**Phase 1**: $\mathcal{A}$ runs an adaptive prediction query, and $\mathcal{C}$ responds to the query.

**Challenge**: $\mathcal{A}$ sends two equal-length plaintext $M_0$ and $M_1$ to $\mathcal{C}$. $\mathcal{C}$ randomly selects a bit $b \in \{0, 1\}$ to obtain ciphertext $\sigma^*$ and returns it to $\mathcal{A}$.

**Phase 2**: $\mathcal{A}$ executes a series of inquiries as in Phase 1, but not allowed to perform extract partial-private key and unsigncryption queries if the user who replaced public key.

**Guess**: A guess bit $b^* \in \{0, 1\}$ is generated by $\mathcal{A}$. $\mathcal{A}$ wins the game if $b^* = b$.

**Definition 1**: Our work satisfies the indistinguishability of chosen multiple identities and the chosen ciphertext attack security (IND-CMID-CCA) if there are no adversaries having a non-negligible advantage to win Game 1.

**Game 2**: SUF-CMID-CCA security

The adversary $\mathcal{A}$ interacts with the challenger $\mathcal{C}$ under the SUF-CMID-CCA security model. We defined the security model as follows:

**Setup**: It is similar to the setup in Game 1.

**Attack**: It is similar to the attack in phase 1 of Game 1.

**Forgery**: $\mathcal{A}$ uses target user set $\{ID_i\}_{i=1,2,\cdots,n}$ and plaintext to forge signatures $\sigma^*$. If any user in the target user set $\{ID_i\}_{i=1,2,\cdots,n}$ unsigncrypt ciphertext $\sigma^*$ correctly, $\mathcal{A}$ wins game. In this process, the ciphertext cannot be obtained by a series of inquiry, and all restrictions are consistent with those in phase 2 of Game 1.

**Definition 2**: Our proposed work can resist the strong unforgeability of chosen multiple identities and ciphertext attack (SUF-CMID-CCA) if there are no adversaries having a non-negligible advantage to win Game 2.

**Game 3**: ANON-CMID-CCA security

The adversary interacts with the challenger under the ANON-CMID-CCA security model. We defined the security model as follows:

**Setup**: $\mathcal{C}$ taking $\lambda$ as input, and returning $Pars$ and $s$ as output, sends $Pars$ to $\mathcal{A}$ and keeps $s$. Then, $\mathcal{A}$ randomly selects identity set $L = \{ID_0, ID_1\}$ to $\mathcal{C}$.

**Phase 1**: This is the same as Game 1.

**Challenge**: $\mathcal{A}$ selects challenge target's identity $L^* = \{ID_2, ID_3, \cdots, ID_n\}$ and plaintext to $\mathcal{C}$. $\mathcal{C}$ randomly select a bit $b \in \{0, 1\}$, formalizes the challenge ciphertext $CT^*$ with a new target identity list $L^* = \{ID_b, ID_2, ID_3, \cdots, ID_n\}$ and sends $CT^*$ to $\mathcal{A}$.

**Phase 2**: It is the same as Game 1.

**Guess**: Finally, a guess bit $b^* \in \{0, 1\}$ is returned by $\mathcal{A}$. $\mathcal{A}$ wins this game if $b^* = b$.

**Definition 3**. Our work satisfies the anonymity of chosen multiple identities and ciphertext attack security (ANON-CMID-CCA) if there are no adversaries having a non-negligible advantage to win Game 3.

## The proposed scheme

We construct a certificateless broadcast signcryption scheme that supports equality test. In the scheme, the broadcaster can signcrypt message for many different receivers, and receiver belonging to the authorization group can unsigncrypt ciphertext to obtain plaintext. Table 1 presents the notions in our proposed scheme. The scheme includes five algorithms:

**Setup**: Inputs security parameter $\lambda$ and returns bilinear pairing $e : G_1 \times G_2 \to G_T$, where $G_1 = \langle P_1 \rangle$, $G_2 = \langle P_2 \rangle$ and they have same the prime order $p$. Define six hash functions as: $H : \{0, 1\}^* \to G_1$, $H_1 : \{0, 1\}^* \times G_1 \times G_1 \to Z_p^*$, $H_2 : \{0, 1\}^* \to Z_p^*$, $H_3 : Z_p^* \to \{0, 1\}^*$, $H_4 : G_T \to G_1$, $H_5 : \{0, 1\}^* \times \{0, 1\}^* \times G_1 \times G_1 \times G_1 \to Z_p^*$ and $f(\cdot) : Z_p^* \to Z_p^*$ is a one-way function. The KGC randomly selects two numbers $s \in Z_p^*$ and $x_T \in Z_p^*$ to compute the public key $PK_{pub} = s \cdot P_1$, $PK_T = x_T \cdot P_2$, master secret key $s$ keeps secret. $Pars = \{P_1, P_2, G_1, G_2, G_T, p, PK_T, PK_{pub}, e, f(), H, H_1, H_2, H_3, H_4, H_5, (E, D)\}$ are system public parameters and $(E(\cdot), D(\cdot))$ is the encryption/decryption algorithm of the Advanced Encryption Standard.

**Keygen**: User and KGC run the keygen algorithm to obtain user's private and public keys.

**Table 1. Notations.**

| Notations | Description |
|---|---|
| $P_1, P_2$ | Generator of group |
| $\lambda$ | Security parameter |
| $H, H_1, H_2, H_3, H_4, H_5$ | Hash functions |
| $Pars$ | System public key |
| $d_i, \omega_i, k, a, r, r_1, \tau$ | Random numbers |
| $(d_i, y_i)$ | Private key |
| $(X_i, Y_i)$ | Public key |
| $z_i$ | Partial private key |
| $M$ | Message |
| $\sigma$ | Ciphertext |

- Set secret-value: A receiver selects number $d_i \in Z_p^*$ at random to act as secret value. It then computes $X_i = d_i \cdot P_1$ and returns $ID_i \parallel X_i$ to the KGC;

- Partial-private key: A receiver submits its identity $ID_i$, master secret key $s$, public key $X_i$ and $Pars$ to KGC. KGC selects a number $\omega_i \in Z_p^*$ and computes $W_i = \omega_i \cdot P_1$, $h_i = H(ID_i)$, $y_i = (\omega_i + h_i \cdot s) \bmod p$, generates user's partial-private key $z_i = y_i + H_1(ID_i, X_i, W_i)$, KGC sends $h_i$, $z_i$ and $W_i$ to user;

- Set private and public key: Computes private key $y_i = z_i - H_1(ID_i, X_i, W_i)$ and public key $Y_i = y_i \cdot P_1$, the partial-private key is true where $y_i \cdot P_1 = W_i + h_i \cdot PK_{pub}$, otherwise, outputs $\perp$. User's public key $PK_i = (X_i, Y_i)$ and private key $SK_i = (d_i, y_i)$.

**Signcryption**: The public parameters $Pars$, message $M$, receiver's identity $ID_i$, sender's private key $SK_s = (d_s, y_s)$, public key $PK_s = (X_s, Y_s)$ and the public key $PK_T$ of TTP are taken as inputs and then performs as follows:

- Constructs $C_i(x) = \prod_{j=1, j \neq i}^{n} \frac{x - x_j}{x_i - x_j} = \sum_{j=0}^{n-1} b_{i,j} x^j \bmod p$, $x_i = H_2(ID_i)$, clearly, $C_i(x_i) = 1$, $C_i(x_j) = 0$, $i \neq j$;

- Selects number $k \in Z_p^*$, $a \in Z_p^*$, $r_1 \in Z_p^*$ and tag $\tau \in Z_p^*$ at random, computes $K = H_3(a)$, $C_0 = (f(M) + f(\tau)) \cdot PK_T$, $C_1 = k \cdot P_2$, $C_2 = e(P_1, r_1 \cdot P_2)^k \cdot a$, $C_3 = E_K(M \parallel \tau)$, $C_4 = e(P_1, P_2)^{f(\tau)}$, $V_i = e(Y_i, C_1)$, $F_i = e(P_1 \cdot H_1(ID_i, X_i, W_i), C_1)$, $T_i = H_4(V_i \cdot F_i) + r_1 \cdot X_i$, $Q_j = \sum_{i=1}^{n} b_{i,j-1} T_i$, $CT = (C_0, C_1, C_2, C_3, C_4, Q_j)$;

- Randomly selects a number $r \in Z_p^*$, computes $R = r \cdot P_1$, $h_s = H_5(M, ID_s, X_s, Y_s, R)$ and $u = d_s + y_s + h_s \cdot r$;

- The final broadcast ciphertext is $\sigma = (CT, u, R)$.

**Unsigncryption**: The broadcast ciphertext $\sigma$, receiver's public key $(X_i, Y_i)$ and receiver's identity $ID_i$ are taken as inputs to perform the following steps:

- Computes $\hat{T}_i = Q_1 + \sum_{j=2}^{n} (x_i^{j-1} \cdot Q_j)$;

- Computes $K' = x_i^{-1} \cdot (\hat{T}_i - H_4(e(P_1 \cdot (y_i + H_1(ID_i, X_i, W_i)), C_1)))$, $K = H_4(C_2 / e(K', C_1))$ and plaintext $M = D_K(C_3)$;

- Checks if $C_0 \stackrel{?}{=} (f(M) + f(\tau)) \cdot PK_T$. Return true if it holds, otherwise, return $\perp$;

- Computes $h_s = H_5(M, ID_s, X_s, Y_s, R)$, if $u \cdot P_1 = X_s + Y_s + h_s \cdot R$, receiver accepts message $(ID_i \parallel M)_{i=1,2,\cdots,n}$, otherwise, outputs $\perp$.

**Equality-test**: Given two ciphertexts $CT = (C_0, C_1, C_2, C_3, C_4, Q_1, Q_2, \cdots, Q_n)$ and $CT' = (C_0', C_1', C_2', C_3', C_4', Q_1', Q_2', \cdots, Q_n')$. TTP checks if $e(C_0 - C_0', P_1)^{x_T^{-1}} = C_4 / C_4'$, returns 1 if it holds and $\perp$ otherwise.

**Correctness**: The correctness of our proposed work as follows:

1. $y_i \cdot P_1 = (\omega_i + h_i \cdot s) \cdot P_1 = W_i + h_i \cdot PK_{pub}$

2. 
$$
\begin{aligned}
\hat{T}_i &= Q_1 + \sum_{j=2}^{n}(x_i^{j-1} \cdot Q_j) \\
&= (b_{1,0}T_1 + b_{2,0}T_2 + \cdots + b_{n,0}T_n) + x_i(b_{1,1}T_1 + b_{2,1}T_2 + \cdots + b_{n,1}T_n) + \cdots \\
&\quad + x_i^{n-1}(b_{1,n-1}T_1 + b_{2,n-1}T_2 + \cdots + b_{n,n-1}T_n) \\
&= (b_{1,0} + x_i b_{1,1} + \cdots + x_i^{n-1}b_{1,n-1})T_1 + (b_{2,0} + x_i b_{2,1} + \cdots + x_i^{n-1}b_{2,n-1})T_2 + \cdots \\
&\quad + (b_{n,0} + x_i b_{n,1} + \cdots + x_i^{n-1}b_{n,n-1})T_n \\
&= C_i(x_i)T_i \\
&= T_i
\end{aligned}
$$

3. 
$$
\begin{aligned}
T_i &= H_4(V_i \cdot F_i) + r_1 \cdot X_i \\
&= H_4(e(Y_i, C_1) \cdot e(P_1 \cdot H_1(ID_i, X_i, W_i), C_1)) + r_1 \cdot x_i P_1 \\
&= H_4(e(y_i \cdot P_1, C_1) \cdot e(P_1 \cdot H_1(ID_i, X_i, W_i), C_1)) + r_1 P_1 \cdot x_i
\end{aligned}
$$

4. $K = H_4(C_2/e(K', C_1)) = H_4(e(P_1, r_1 \cdot P_2)^k \cdot (a)/e(r_1 \cdot P_1, k \cdot P_2)) = H_4(a)$

5. $u \cdot P_1 = X_s + Y_s + h_s \cdot R = d_c \cdot P_1 + y_s \cdot P_1 + h_s \cdot rP_1 = u \cdot P_1$

6. 
$$
\begin{aligned}
e(C_0 - C_0', P_1)^{x_T^{-1}} &= e(C_0 - C_0', x_T^{-1} \cdot P_1) \\
&= \frac{e(f(M) + f(\tau) \cdot x_T \cdot P_2, x_T^{-1} \cdot P_1)}{e(f(M) + f(\tau') \cdot x_T \cdot P_2, x_T^{-1} \cdot P_1)} \\
&= \frac{e(f(M) + f(\tau) \cdot P_2, P_1)}{e(f(M) + f(\tau') \cdot P_2, P_1)} \\
&= \frac{e(f(M) \cdot P_2, P_1)e(f(\tau) \cdot P_2, P_1)}{e(f(M) \cdot P_2, P_1)e(f(\tau') \cdot P_2, P_1)} \\
&= \frac{e(P_2, P_1)^{f(\tau)}}{e(P_2, P_1)^{f(\tau')}} \\
&= \frac{C_4}{C_4'}
\end{aligned}
$$

## Security proof

We proof the security of our work under the hard problem and security model in Section 4.

**Theorem 1**. If the BDH problem in $(G_1, G_2)$ is hard, our work is secure against the IND-CMID-CCA of the $\mathcal{A}_I$.

**Proof**: Simulator $\mathcal{B}$ is created to solve the hard problems of BDH in $(G_1, G_2)$. Inputs $(P_1, \beta_1 P_1, \beta_2 P_1, V)$ of the DDH problem and checks $V \overset{?}{=} \beta_1 \beta_2 P_1$. $\mathcal{B}$ and $\mathcal{A}_I$ to simulates the security game.

**Setup**: $\mathcal{B}$ sets system public key $PK_{pub} = \alpha P_1 = \varphi(\alpha P_2)$ and $PK_T = \beta_1 P_1$, and then sends system parameter $Pars = \{P_1, P_2, G_1, G_2, p, e, H, H_1, H_2, H_3, H_4, H_5, (E, D)\}$ to $\mathcal{A}_I$. After receiving $Pars$, $\mathcal{A}_I$ outputs the target identity $\{ID_i\}_{i=1,2,\cdots,n}$.

**Phase 1**: $\mathcal{B}$ sets $H, \{H_i\}_{i=1,2,\cdots,5}$ and runs a series of queries, returns the results to $\mathcal{A}_I$ and the query results are stores in lists $H, H_1, H_2, H_3, H_4, H_5$.

**H query**: After receiving the query from the adversary on the target identity $\{ID_i\}_{i=1,2,\cdots,n}$, $\mathcal{B}$ creates $(ID_i, W_i, \psi_i, \theta_i)$ in the list $H$ and initializes it to null. If the identity exists in the tuple, returns $W_i$. Otherwise, randomly selects a bit $\psi_i \in \{0, 1\}$ and an integer $\theta_i \in Z_p$. If $\psi_i = 0$,

computes $W_i = \theta_i \cdot P_1$, otherwise, sets $W_i = \theta_i \cdot bP_1 = \theta_i \cdot \varphi(bP_2)$ and adds $(ID_i, W_i, \psi_i, \theta_i)$ to the list $H$. Finally, $\mathcal{B}$ returns $W_i$ to $\mathcal{A}_i$;

$H_1$ **query**: Inputs $(ID_i, X_i, W_i)$, $\mathcal{B}$ runs $H_1$ query, checks whether $(ID_i, X_i, W_i, d_i)$ exists in the list $H_1$. If it does, returns $d_i$ to $\mathcal{A}_I$. Otherwise, $\mathcal{B}$ randomly selects $d_i \in Z_p^*$ to sends it to $\mathcal{A}_I$ and stores $(ID_i, X_i, W_i, d_i)$ in the list $H_1$;

$H_2$ **query**: A list $L$ is created and initialized to empty. If the identity in the $(ID_i, x_i)$ query already exists in the list, $\mathcal{B}$ returns $H_2(ID_i) = x_i$. Otherwise, randomly selects an integer $x_i \in Z_p^*$ sends to $\mathcal{A}_I$ and adds $(ID_i, x_i)$ to the list $L$. Finally, it returns $x_i$ to $\mathcal{A}_I$;

$H_3$ **query**: The identity $ID_i$ is taken as input. $\mathcal{B}$ creates a tuple $(ID_i, k_i)$ in the list $H_3$ and initializes it to empty. If $(ID_i, k_i)$ exists in the list $H_3$, it will be returned $k_i$ to $\mathcal{A}_I$. Otherwise, randomly selects $k_i \in Z_p$ and returns to the adversary and add it to the tuple $(ID_i, k_i)$ of the list $H_3$;

$H_4$ **query**: Inputs the identity $ID_i$, $\mathcal{B}$ creates $(ID_i, T_i)$ in the list $H_4$ and initializes it to empty. If $(ID_i, T_i)$ exists in the list $H_4$, it returns $T_i$ to $\mathcal{A}_I$. Otherwise, randomly selects an integer $T_i \in G_1$ and returns to $\mathcal{A}_I$ and adds $(ID_i, T_i)$ of the list $H_4$;

$H_5$ **query**: $(M, ID_i, X_i, Y_i, R)$ is taken as input. If $(M, ID_i, X_i, Y_i, R, h_k)$ exists in the list $H_5$, $\mathcal{B}$ send $h_k$ to $\mathcal{A}_I$, otherwise, randomly selects $h_k \in Z_p^*$ and sends it to $\mathcal{A}_I$ and store $(M, ID_i, X_i, Y_i, R, h_k)$ in the list $H_5$;

**Key query**: If $(ID_i, SK_i, PK_i, x_i, z_i)$ exists in the list $H_i$, keep tuple $(ID_i, SK_i, PK_i, x_i, z_i)$. Otherwise, $\mathcal{B}$ responds as follows:

- If $ID_i = ID_j$, $\mathcal{B}$ randomly selects $x_i, w_i \in Z_p^*$, computes $X_i = x_i \cdot P_1$, $W_i = \omega_i \cdot P_1$, $y_i = \omega_i + H_1$ $(ID_i, X_i, W_i) \cdot s \bmod p$, $Y_i = y_i \cdot P_1$, $PK_i = (X_i, Y_i)$ and updates tuple $(ID_i, SK_i, PK_i, x_i, z_i)$ in the list $H_i$ and the tuple $(ID_i, X_i, W_i, d_i)$ in the list $H_1$ respectively;

- If $ID_i \neq ID_j$, $\mathcal{B}$ randomly selects $x_i, y_i \in Z_p^*$, computes $X_i = x_i \cdot P_1$, $Y_i = y_i \cdot P_1$, $PK_i = (X_i, Y_i)$, $SK_i = (d_i, y_i)$ and updates tuple $(ID_i, SK_i, PK_i, x_i, z_i)$ in list $H_i$ and tuple $(ID_i, X_i, W_i, d_i)$ in list $H_1$ respectively.

**Secret-value query**: After receiving the request from $\mathcal{A}_I$, it sends $x_i$ to $\mathcal{A}_I$ if $(ID_i, SK_i, PK_i, x_i, z_i)$ exists in list $H_i$.

**Extract Partial-private key query**: Inputs $ID_i$, $\mathcal{B}$ executes as follows:

- If $ID_i = ID_j$, $\mathcal{B}$ sends $\perp$ to $\mathcal{A}_I$;

- If $ID_i \neq ID_j$, and if $(ID_i, SK_i, PK_i, w_i, x_i, z_i)$ exists in the list $H_i$, $\mathcal{B}$ sends partial-private key $z_i$ to $\mathcal{A}_I$, otherwise, runs key query and returns tuple $(ID_i, SK_i, PK_i, x_i, z_i)$ and sends partial-private key $z_i$ to $\mathcal{A}_I$.

**Private query**: After receiving the request from $\mathcal{A}_I$, $\mathcal{B}$ sends public key $PK_i$ to $\mathcal{A}_I$ if $(ID_i, SK_i, PK_i, x_i, z_i)$ exists in list $H_i$, otherwise, $\mathcal{B}$ runs key query, return $(ID_i, SK_i, PK_i, x_i, z_i)$ and sends public key $PK_i$ to $\mathcal{A}_I$ and responds as follows:

- If $ID_i = ID_j$, $\mathcal{B}$ sends $\perp$ to $\mathcal{A}_I$;

- If $ID_i \neq ID_j$, if $(ID_i, SK_i, PK_i, w_i, x_i, z_i)$ in exists in the list $H_i$, $\mathcal{B}$ sends private key $SK_i$ to $\mathcal{A}_I$, otherwise, $\mathcal{B}$ runs key query, return tuple $(ID_i, SK_i, PK_i, x_i, z_i)$ and sends $SK_i$ to $\mathcal{A}_I$.

**Public key replace query**: If tuple $(ID_i, SK_i, PK_i, w_i, x_i, z_i)$ exists in the list $H_i$ after receiving the request, $\mathcal{B}$ replaces $PK_i$ with public key $PK_i'$, otherwise, $\mathcal{B}$ will be stored tuple $(ID_i, SK_i, PK_i, x_i, z_i)$ in list $H_i$.

**Signcryption query**: If $ID_i \neq ID_j$, $i = 1, 2, \cdots, n$, $\mathcal{B}$ runs the private key query, output $SK_s$, ciphertext $CT$, and sends $CT$ to $\mathcal{A}_I$, otherwise, $\mathcal{B}$ respond as follows:

- Selects $ID_i$, computes $x_i = H_2(ID_i)$;

- Constructs $C_i(x) = \prod\limits_{j=1, j\neq i}^{n} \frac{x-x_j}{x_i-x_j} = \sum\limits_{j=0}^{n-1} b_{i,j} x^j \bmod p$;

- Randomly selects the integers $k \in Z_p^*$, $A \in Z_p^*$ and $\tau \in Z_p^*$, computes $C_1 = k \cdot P_2$, $K = H_3(A)$ and $C_3 = E_K(M \parallel \tau)$;

- Randomly selects $r \in Z_p^*$, computes $R = r \cdot P_1$, $h_s = H_5(M, ID_s, X_s, Y_s, R)$ and $u = d_s + y_s + h_s \cdot r$;

- Outputs $CT = (C_0, C_1, C_2, C_3, C_4, Q_t)$ to $\mathcal{A}_I$.

**Unsigncryption query**: The adversary requires $\mathcal{B}$ to run unsigncryption query on ciphertext $CT$ and identity $ID_i$. After receiving the request, if $ID_i = ID_j$, $i = 1, 2, \cdots, n$, $\mathcal{B}$ sends $\perp$ to $\mathcal{A}_I$, otherwise, $\mathcal{B}$ responds as follows:

- Inputs the broadcaster identity $ID_s$, authorized device identity $ID_i$ and ciphertext $CT$, and then computes $W = r_1 \cdot P_1$;

- Computes $K' = H_4(C_2/e(W, C_1))$;

- If $i = 1$ in the $H_4$ query, $\mathcal{B}$ recovers the plaintext $M \parallel ID_i = D(K', C_3)$ from the list $H_4$, uses the symmetric key $K'$ and returns it to $\mathcal{A}_I$. Checks if $C_0 = f(M) \cdot PK_T + f(\tau) \cdot PK_T$. If so, the execution is completed. If $i$ does not exceed the number of $H_4$ queries, $\mathcal{B}$ returns $M$ to $\mathcal{A}_I$, otherwise, outputs $\perp$;

- If $R = h_i \cdot P_2 - u \cdot Y_i$, $\mathcal{B}$ returns $M$ to $\mathcal{A}$, otherwise, outputs $\perp$.

**Challenge**: $\mathcal{A}_I$ selects two equal-length $M_0$, $M_1$ and challenger identity and public key set $S^* = (ID_1/X_1, \cdots, ID_l/X_l)$. In phase 1, $\mathcal{A}_I$ cannot uses the identity $ID_i \in S^*$ to runs the private key query. If $\psi_i = 1$ at the tuple $(ID_i, W_i, \psi_i, \theta_i)$ in list $H_1$, $\mathcal{B}$ responds as follows:

- Sets $C_1^* = k \cdot P_2$;

- Computes $x_i^* = H_2(ID_i)$, constructs $C_i(x) = \prod\limits_{j=1, j\neq i}^{n} \frac{x-x_j^*}{x_i^*-x_j^*} = \sum\limits_{j=0}^{l} b_{i,j} x^j$;

- Randomly selects $r_1 \in Z_p$, $R_i \in G_1$, computes $T_j = R_i + r_1 \cdot X_j$ and $Q_t = \sum\limits_{j=1}^{n} b_{j,t-1} T_j$;

- Randomly selects $A \in G_T$ and $\tau \in \{0, 1\}^t$, computes $K = H_3(A)$, $C_2^* = e(P_1, C_1^*)^{r_1} \cdot A$ and $C_3^* = E_K(M_\beta \parallel \tau), \beta \in \{0, 1\}$;

- Computes $C_0^* = f(M_\beta) \cdot PK_T + V$ and $C_4^* = e(\alpha P_1, P_2)$;

- Sends $CT^* = (C_0^*, C_1^*, C_2^*, C_3^*, C_4^*, Q_t^*)$ to $\mathcal{A}_I$.

**Phase 2**: $\mathcal{A}_I$ runs a series of adaptive queries consistent with those in phase 1, but challenge ciphertext $CT^*$ cannot be decrypted. If $ID_i \in S^*$, extract partial-private key query is not allowed.

**Guess**: Given a bit $\beta' \in \{0, 1\}$, if $V = \beta_1\beta_2 P_1$, $CT^*$ is valid. Suppose $\mathcal{A}_I$ must run $H_5$ query as $C_1^* = c \cdot P_2$, $H(ID_i) = q_i \cdot b \cdot P_1$ and $PK_{pub} = aP_1$, $\mathcal{B}$ computes $e(P_1, P_2)^{abc} = (X_i) q_i^{-1}$. Therefore, our scheme is secure against the IND-CMID-CCA.

**Theorem 2**. If the DDH problem in $G_1$ is hard, our proposed scheme is secure against the IND-CMID-CCA of $\mathcal{A}_{II}$.

**Proof**: Simulator $\mathcal{B}$ is created to solve the hard problems of DDH in $G_1$, Let $P_1$, $aP_1$, $bP_1$, $W \in G_1$, where $a, b \in Z_p^*$ are unknown, judge whether $W = abP_1$. $\mathcal{B}$ and $\mathcal{A}_{II}$ to simulates the security game.

**Setup**: sets system public key $PK_{pub} = \alpha P_1$ and $PK_T = \beta P_1$, and then sends system parameter $Pars = \{P_1, P_2, G_1, G_2, p, e, H, H_1, H_2, H_3, H_4, H_5, (E, D)\}$ to $\mathcal{A}_{II}$.

**Phase 1**: $\mathcal{A}_{II}$ adaptively issue a series of queries.

**H query**: After receiving the query from the adversary on the target identity $\{ID_i\}_{i=1,2,\cdots,n}$, $\mathcal{B}$ creates $(ID_i, W_i)$ in the list $H$ and initializes it to null. If the identity exists in the tuple, returns $W_i$. Otherwise, randomly selects an integer $\theta_i \in Z_p$, computes $W_i = \theta_i \cdot P_1$ and adds $(ID_i, W_i, \psi_i)$ to the list $H_1$. Finally, $\mathcal{B}$ returns $W_i$ to $\mathcal{A}_{II}$.

**$H_1,H_2,H_3,H_4,H_5$ query**: It is similar to the $H_1,H_2,H_3,H_4,H_5$ query in Theorem 1.

**Public key query**: After receiving the query from the adversary on the target identity $\{ID_i\}_{i=1,2,\cdots,n}$, if the $(ID_i, W_i, x_i)$ has existed in the list $H$ and initializes it to null. If the identity exists in the tuple, returns $W_i$. Otherwise, randomly selects a bit $\psi_i \in \{0, 1\}$ and an integer $a_i \in Z_p$. If $\psi_i = 0$, computes $X_i = a_i \cdot P_1$, otherwise, sets $X_i = a_i \cdot bP_1$ and adds $(ID_i, a_i, X_i, \psi_i)$ to the list $H$. Finally, $\mathcal{B}$ returns $X_i$ to $\mathcal{A}_{II}$.

**Unsigncryption query**: It is similar to the Unsigncryption query in Theorem 1.

**Challenge**: It is similar to the Challenge in Theorem 1.

**Phase 2**: $\mathcal{A}_{II}$ runs a series of adaptive queries consistent with those in phase 1, but challenge ciphertext $CT^*$ cannot be decrypted.

**Guess**: Given its guess $\beta$, if $\beta = \beta'$, $\mathcal{A}_{II}$ wins the game with non-ignorable advantage. When $W = abP_1$, $CT^* = (C_0^*, C_1^*, C_2^*, C_3^*, C_4^*, Q_t^*)$ is valid since $T_i = H_4(V_i \cdot F_i) + r_1 \cdot W = H_4(V_i \cdot F_i) + r_1 \cdot (abP_1)$, $C_4 = e(P_1, P_2)^{f(\tau)} = e(P_1, P_2)^{f(b)}$. This means that $r_1 = a$, $\tau = b$ in the signcryption. Thus, if $\mathcal{A}_{II}$ breaks the proposed work, $\mathcal{B}$ is able to solve the DDH problem.

**Theorem 3**. Define one-way functions $H$ and $\{H_i\}_{i=1,2,\cdots,5}$. If the CDH problem in $(G_1, G_2)$ is hard, the scheme is secure against the SUF-CMID-CCA of $\mathcal{A}_I/\mathcal{A}_{II}$.

**Proof**: The simulator $\mathcal{B}$ is created to solve CDH problem in $(G_1, G_2)$, $\mathcal{B}$ interacts with $\mathcal{A}$ as follows:

**Setup**: It is similar to the setup in Theorem 1.

**Phase 1**: It is similar to Phase 1 in Theorem 1.

**Public key query**: If the challenge identity $ID_i^*$ is received, $\mathcal{B}$ will sends $PK_{pub} = a \cdot P_1$ to $\mathcal{A}_I$, otherwise, randomly selects $x_i \in Z_p^*$ and computes $PK_{pub} = x_i \cdot P_1$.

**Private key query**: If the challenge identity $ID_i^*$ is received, $\mathcal{B}$ return $\perp$, otherwise, run private key query and sends $x_i \in Z_p^*$ to $\mathcal{A}$.

**Signcryption query**: $\mathcal{B}$ runs the following steps on the identity:

- If $ID_i = ID_i^*$, $\mathcal{B}$ randomly selects $a_i, b_i \in Z_p^*$, computes $R = a_i \cdot P_1 - b_i \cdot Y_i$;

- If $ID_i \neq ID_i^*$, $\mathcal{B}$ generates ciphertext to $ID_i = ID_i^*$.

**Forge**: If $ID_i = ID_i^*$, $\mathcal{A}$ uses the identity set $L = \{ID_1, ID_2, \cdots, ID_n\}$ to forge a signature $\sigma^* = (CT^*, u^*)$. If $\sigma P_1 = X_i + Y_i + h_i \cdot R$ holds, $\mathcal{A}$ wins the game, defines $PK_i' = y_i^{-1} \cdot PK_i$ and $r \cdot X_i = y_i(PK_i + PK_{pub})$, $\mathcal{B}$ computes $r \cdot X_i = PK_i' + \alpha\beta P$ and outputs $\alpha\beta P = r \cdot X_i - PK_i$, otherwise, outputs the terminator $\perp$.

**Theorem 4**. Define one-way functions $H$ and $\{H_i\}_{i=1,2,\cdots,5}$. If the DBDH problem in $(G_1, G_2)$ is hard, our work is secure against the ANON-CMID-CCA of $\mathcal{A}_I$.

**Proof**: Simulator $\mathcal{B}$ is created to solve DBDH problem in $(G_1, G_2)$. $\mathcal{B}$ interacts with $\mathcal{A}_I$ as follows:

**Setup**: The simulator sets system public key $PK_{pub} = aP_1$, sends system parameter $Pars = \{G_1, G_2, G_T, p, e, f(), H, H_1, H_2, H_3, H_4, H_5, (E, D)\}$ to $\mathcal{A}_I$, $\mathcal{A}_I$ outputs the target identity $\{ID_i\}_{i=1,2,\cdots,n}$.

**Phase 1**: $\mathcal{A}_I$ executes a series of adaptive queries consistent with Theorem 1.

**Challenge**: $\mathcal{A}_I$ cannot run partial private key query on $ID_i \in \{S_0^*, S_1^*\}$. Message $M$, two identities and public key sets with different lengths $S_0^* = (ID_0^*/X_0^*, ID_2/X_2, \cdots, ID_l/X_l)$ and $S_1^* = (ID_1^*/X_1^*, ID_2/X_2, \cdots, ID_l/X_l)$ are taken as inputs. $\mathcal{B}$ runs as follows:

- Sets $C_1^* = k \cdot P_2$;

- Retrieves $(ID_\beta^*, W_\beta^*, \psi_\beta^*, \theta_\beta^*)$ on the identity $ID_\beta^*$, if $\psi_\beta^* = 0$, outputs $\perp$, if $\psi_\beta^* = 1$, sets $W_\beta^* = \theta_\beta^* \cdot bP_1$ and $X_\beta^* = Z^{\theta_\beta^*}$. Then, $T_\beta^*$ is obtained by $H_3$ query;

- Computes $x_\beta^* = H_3(ID_\beta^*)$, $x_i^* = H_2(ID_i)$, $i \in \{2, 3, \cdots, l\}$;

- Constructs $C_i(x) = \prod\limits_{j=1,j\neq i}^{n} \frac{1}{x_i^* - x_j^*} \cdot \left(x - x_j^*\right) = \sum\limits_{j=0}^{l} b_{ij}x^j$, $i \in \{2, 3, \cdots, l\}$;

- Randomly selects integer $r_1 \in Z_p^*$, $R_i \in G_1$, $i \in \{2, 3, \cdots, l\}$, computes $T_j = R_i + r_1 \cdot X_j$ and $T_\beta = R_\beta + r_1 \cdot X_\beta^*$;

- Computes $Q_i = \sum\limits_{j=1}^{l} a_{j,i-1} T_j$, $i \in \{\beta, 2, 3, \cdots, l\}$;

- Randomly selects integers $A \in G_p$ and $\tau \in Z_p$, computes $C_2^* = e(P_1, C_1^*)^{r_1}$, $C_3^* = E_K(M_\beta\|\tau)$, $\beta \in \{0, 1\}$;

- Computes $C_0^* = f(M_\beta + \tau) \cdot PK_T$ and $C_4^* = e(P_1, P_2)^\tau$;

- Sends challenge ciphertext $CT^* = (C_0^*, C_1^*, C_2^*, C_3^*, C_4^*, Q_t^*)$ to $\mathcal{A}_I$.

**Phase 2**: $\mathcal{A}_I$ runs adaptive queries consistent with those of phase 1, but is not allowed to perform partial-private key, public key replace, unsigncryption query for identity $ID \in \{ID_0^*, ID_1^*\}$.

**Guess**: $\mathcal{A}_I$ guesses $b'$, if $b' = b$, $\mathcal{B}$ outputs 1, otherwise, outputs 0.

**Analysis**: Simulator $\mathcal{B}$ is indistinguishable from the scheme in the above game. When $Z = e(P_1, P_2)^{\alpha\beta c}$, assuming that $k^* = c$. $C_3^* = E_K(M_\beta\|\tau)$ where $K = H_3(A)$ is a random element. Therefore, $\mathcal{A}_I$ view $M_\beta$ as independent, and our work is secure against the ANON-CMID-CCA.

**Theorem 5**. If the DDH problem in $G_1$ is hard, our proposed work is secure against the ANON-CMID-CCA of $\mathcal{A}_{II}$.

**Proof**: Simulator $\mathcal{B}$ is created to solve the hard problems of DDH in $G_1$, Let $P_1, aP_1, bP_1, W \in G_1$, where $a, b \in Z_p^*$ are unknown, judge whether $W = abP_1$. $\mathcal{B}$ and $\mathcal{A}_{II}$ to simulates the security game.

**Setup**: It is similar to the Setup in Theorem 2.

**Phase 1**: It is similar to the Phase 1 in Theorem 2.

**Challenge**: It is similar to the Challenge in Theorem 2.

**Phase 2**: $\mathcal{A}_{II}$ is not allowed performed public key query and unsigncryption query with $ID$, where $ID \in \{ID_0^*, ID_1^*\}$.

**Guess**: Given $\beta'$, if $\beta = \beta'$, outputs 1, otherwise, outputs 0.

**Analysis**: Simulator $\mathcal{B}$ is indistinguishable from the scheme in the above game. When $W = abP_1$, assuming that $r_1 = a$. In addition to $W$ is a random element of group $G_1$, $C_3^* = E_K(M_\beta\|\tau)$

where $K = H_3(A)$ is a random element. Therefore, $\mathcal{A}_{II}$ view $M_\beta$ as independent, and our work is secure against the ANON-CMID-CCA.

## Performance evaluation

### Functional comparison

We evaluate the functions of the proposed work with those of five existing broadcast signcryption schemes [15, 22–25]. From Table 2, scheme [15] outsource verification operation to gateway, which reduces the computation cost of the receiver at decryption stage. However, there is the problem of key escrow. Scheme [22] presents a multi-signcryption scheme with public verifiability to reduce the threats of private key escrow and replay attack but can't eliminate duplicate copies in the system. The scheme [23] propose the certificateless broadcast signcryption scheme, but can't ensure the privacy of the receiver. Scheme [24] set a smaller key unit based-identity signcryption, which is not applicable to equipment with limited resources, and there is the risk of the receiver's privacy leaking. The scheme [25] design a user access control scheme which fails to achieve receiver's privacy preservation and the computation cost of unsigncryption is higher than the proposed work.

### Efficiency analysis

We compare the computation times of our work with those of the existing schemes [15, 22] as shown in Table 3. The communication cost between our work and other schemes is shown in Table 4.

**1) Computation cost**. We compare the computation times of our work and existing schemes [15, 22] is shown in Table 3. $T_e$, $T_m$, $T_p$, $T_h$, $T_{Inv}$ represents the time of executing

**Table 2. Comparison of functions.**

| The scheme | Certificateless | Broadcast | Privacy-preserving | De-duplication |
|---|---|---|---|---|
| Ref [15] | No | Yes | Yes | No |
| Ref [22] | No | No | Yes | Yes |
| Ref [23] | Yes | No | No | No |
| Ref [24] | No | No | Yes | Yes |
| Ref [25] | Yes | No | No | No |
| Ours | Yes | Yes | Yes | Yes |

**Table 3. Computation costs.**

| The scheme | Signcryption | Unsigncryption |
|---|---|---|
| Ref [15] | $(2n + 1)T_m + (4n + 1)T_h + T_{Inv} + (n - 1)T_e$ | $2(n + 1)T_m + (3n + 1)T_h + (n - 1)T_e$ |
| Ref [22] | $2(n + 2)T_m + T_p + 4T_h$ | $2T_h + 3T_p$ |
| Ours | $(2n + 3)T_m + 3T_h + nT_e + 4T_p$ | $(2 + n)T_m + 4T_h + (n - 1)T_e$ |

**Table 4. Communication cost.**

| The scheme | Ciphertext size |
|---|---|
| Ref [15] | $16n + 48$ bytes |
| Ref [22] | $32n + 16$ bytes |
| Ours | $16n + 96$ bytes |

exponential, multiplication, bilinear pairing, hash, and multiplication inversion operation, respectively. The operation time sequence is $T_p > T_e > T_m > T_h > T_{Inv}$. $n$ represents the number of users. The computation cost increases as $n$ grows.

**2) Communication cost**. We compare the communication costs of the proposed work with those of schemes [15, 22] in shown Table 4. We set $|Z_p^*|$=16 bytes and $|G|$=32 bytes. $n$ represents the number of users. The ciphertext size are $n|Z_p^*| + |Z_p^*| + |G|$, $n|G| + |Z_p^*|$, $n|Z_p^*| + |G| + 4|Z_p^*|$ in [15, 22] and our scheme, respectively. The communication cost grows linearly with $n$ from Table 4.

## Experimental analysis

The experiment is using bilinear pairing-based cryptography library under the Linux operating system. The parameter type of bilinear pairing package is Type-A. It uses the C programming language and is programmed and executed on 2.60 GHz CPU and 8 GB RAM PC. We compare the computation time of [15, 22] and our proposed scheme of signcryption and unsigncryption algorithms, and set the number of devices from the smart grid at 10, 20, 30, 40, 50, 60, 70 and 80, respectively. The number of devices on the smart grid can dynamically adjusted to manage authorized devices more flexibly.

As is shown in Fig 3. that the computation time of our work in data signcryption stage is lower than scheme [22]. Although computation efficiency of scheme [15] is higher than our scheme, the proposed work has higher security and practical application value. It is also

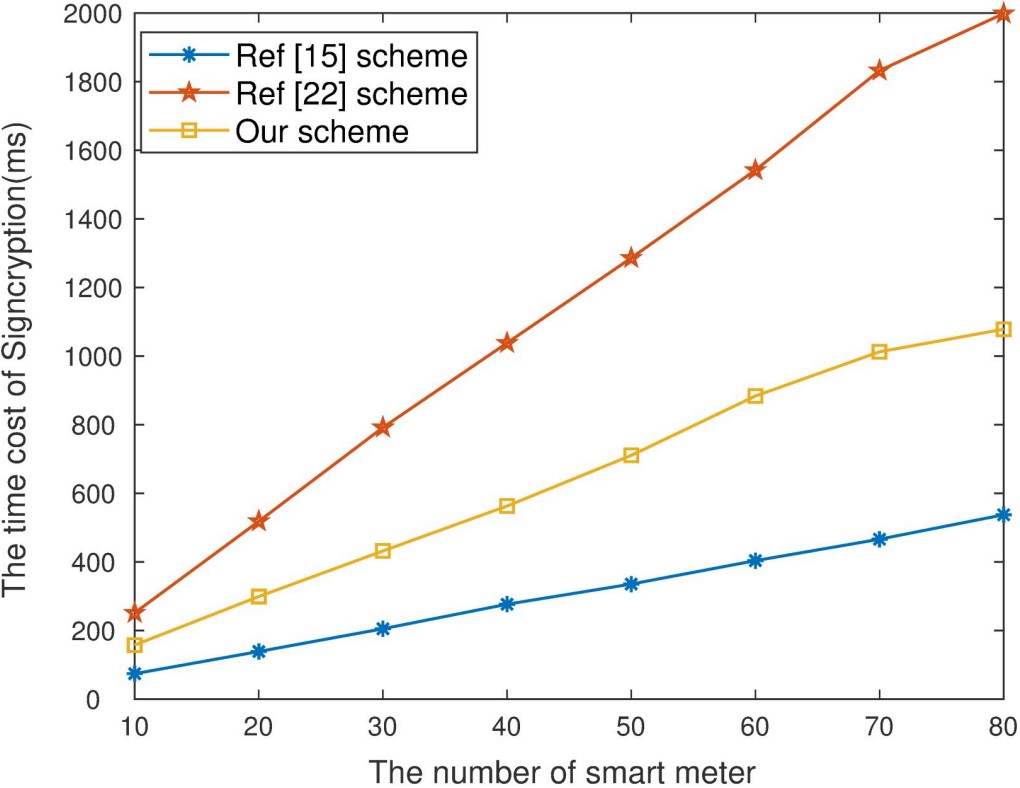

**Fig 3. Time cost of signcryption.**

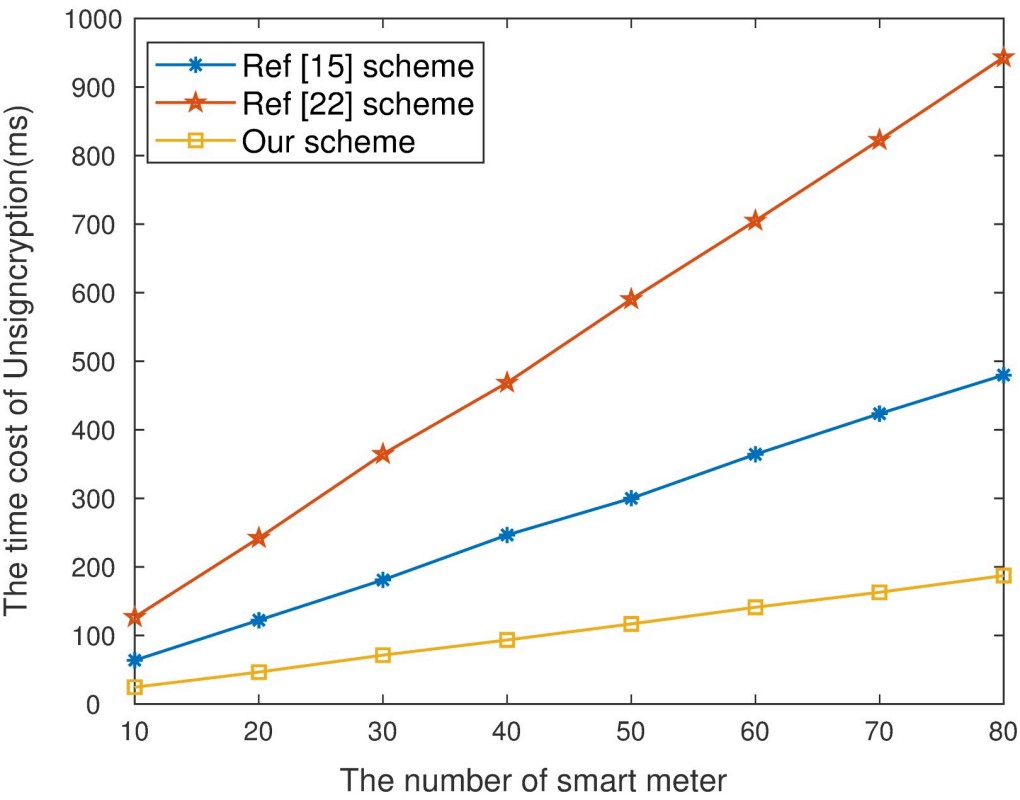

**Fig 4. Time cost of unsigncryption.**

concluded from Fig 4. that the computation efficiency of our work in the data unsigncryption stage is higher than existing schemes [15, 22]. When the number of devices is 20, the computation time of our scheme is 46.565ms, [15, 22] are 122.278ms and 242.153ms respectively. Fig 5. show that the communication costs of our work are lower than [22]. The computation of unsigncryption and communication cost of [22] is highly than our proposed scheme. The core reason is equality test cannot be performed. Although communication costs of [15] is higher than our scheme, our proposed work has higher security and can better guarantee the privacy of users.

## Conclusion

Currently, there exist malicious attackers in the smart grid, causing the smart grid to face some security threats, such as user forging smart meter data, unauthorized user access to sensitive information leading to privacy leakage, and so on. To realize the private preservation of smart meter's identities and the confidentiality of sensitive information, guarantee the security of data communication and solve the problem of insufficient transmission network bandwidth resources, we construct a broadcast signcryption scheme supporting equality test based on certificateless cryptosystem. The scheme realizes the anonymity between receivers and ensures the privacy of data. In addition, our work also achieves data deletion function of the same ciphertext, which greatly saves the network bandwidth and ciphertext storage space. Finally, an analysis of the existing broadcast signcryption schemes and our proposed scheme reveals that our proposed work has higher practical application value.

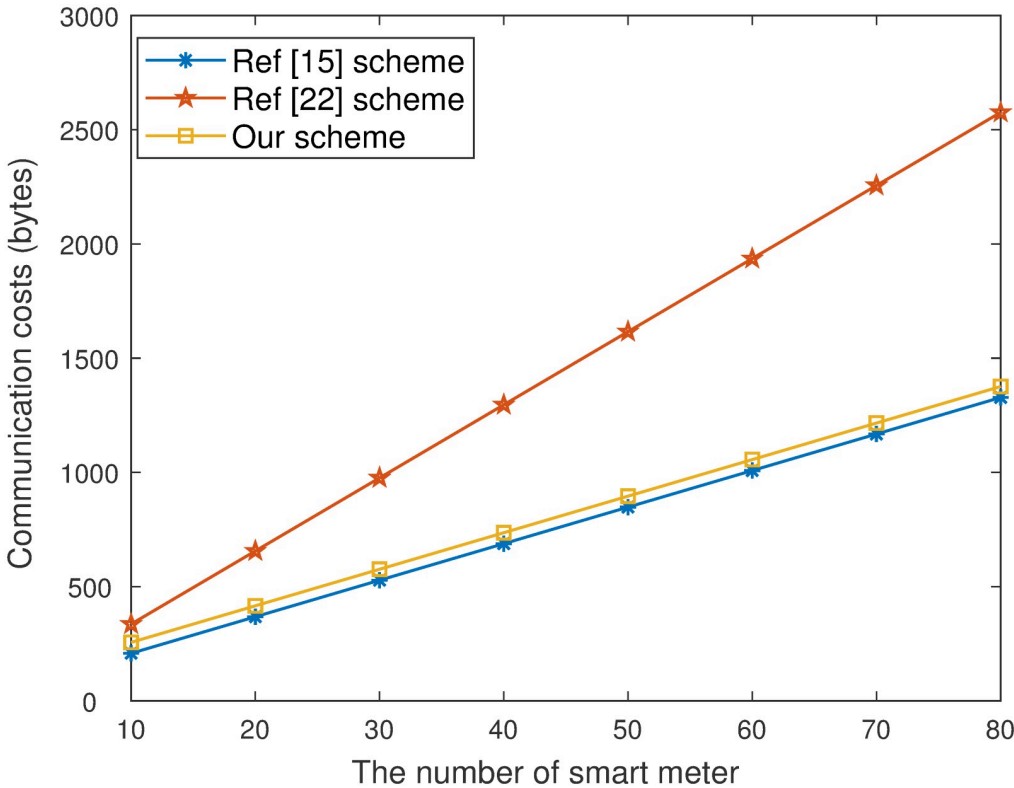

**Fig 5. Communication costs.**

## Author Contributions

**Writing – original draft:** Shufen Niu, Runyuan Dong, Lizhi Fang.

**Writing – review & editing:** Shufen Niu, Runyuan Dong, Lizhi Fang.

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
