## [Decision Letter · Decision Letter 0]

10 Apr 2023

PONE-D-23-08826Certificateless broadcast signcryption scheme supporting equality test in smart gridPLOS ONE

Dear Dr. Dong,

Thank you for submitting your manuscript to PLOS ONE. After careful consideration, we feel that it has merit but does not fully meet PLOS ONE’s publication criteria as it currently stands. Therefore, we invite you to submit a revised version of the manuscript that addresses the points raised during the review process.

We look forward to receiving your revised manuscript.

Kind regards,

Pandi Vijayakumar, Ph.D

Academic Editor

PLOS ONE

Journal Requirements:

   "This work was supported by the National Natural Science Foundation of China (No.62241207, No.61662060); Gansu Science and Technology Program (22JR5RA158), Industrial support plan project of Gansu Provincial Department of Education (2022CYZC17). The authors gratefully acknowledgement the anonymous reviewers for their detailedsuggestions."

   " The author(s) received no specific funding for this work."

Additional Editor Comments:

The paper has many serious issues. The conclusion section is missing, motivation is missing and the paper needs to be rewritten.

Reviewers' comments:

Reviewer's Responses to Questions

**Comments to the Author**

1. Is the manuscript technically sound, and do the data support the conclusions?

Reviewer #1: Yes

Reviewer #2: Yes

2. Has the statistical analysis been performed appropriately and rigorously? 

Reviewer #1: Yes

Reviewer #2: Yes

3. Have the authors made all data underlying the findings in their manuscript fully available?

Reviewer #1: Yes

Reviewer #2: Yes

4. Is the manuscript presented in an intelligible fashion and written in standard English?

Reviewer #1: Yes

Reviewer #2: Yes

5. Review Comments to the Author

Reviewer #1: The authors seek to propose a broadcast signcryption scheme supporting equality test based on certificateless Cryptosystem for smart grid. I have the following suggestions that need to be answered.

1) In the abstract, the sentence needs to be revised. “However, the current scheme cannot ensure user privacy or efficient network bandwidth utilization.”

2) I would suggest the authors to add a detailed motivation for the proposed scheme.

3) A detailed paragraph regarding the hardness efficiency details of the proposed scheme needs to be added above the contribution’s headings. The authors need to show why the proposed scheme is efficient and how.

4) The entire literature review section is written in the present tense. The words like constructed, presented, and proposed need to be changed to propose, construct and present.

5) The related work is limited; I would suggest the authors expand the related work. Besides, I would suggest the authors to conclude the related work. Currently, the authors did not conclude the related work; what did they learn after reviewing the literature?

6) In the system model, the authors just define the terminology used in the scheme; however, the workflow of the proposed scheme is missing.

7) The scheme's definition needs to be named with the subsection, like System Definition.

8) I haven’t seen any details in the experimental analysis for the number of devices. How and from where did the authors add the details for the number of devices? How can the authors set the devices hypothetically?

The authors did not conclude their research; a conclusion is necessary and must be added.

9) The article needs to be thoroughly proofread to remove all the grammatical and typos.

Reviewer #2: The technical aspects of the paper are fine. The authors, however, will need to make some minor adjustments before publishing it. The following are some of the concerns that the authors should address:

1. The introduction is quite brief. The authors should add to it, highlighting the smart grid's vulnerabilities and the merits of using a certificateless signcryption scheme.

2. Once more, the literature review section is extremely brief. Few relevant articles precisely on the same subject are ignored, for example (https://doi.org/10.3390/su131910891). The authors must contribute a few additional articles to this section.

3. I could not locate the section's conclusion. I have no idea why the conclusion is missing. Authors must provide justification.

4.  The authors need to revise the article with correct usage of English, grammatical mistakes, and punctuation errors.

6. PLOS authors have the option to publish the peer review history of their article (what does this mean?). If published, this will include your full peer review and any attached files.

Reviewer #1: No

Reviewer #2: No

---

## [Author Response · Author response to Decision Letter 0]

3 Jul 2023

Comments of editor

Comments to the author

The paper has many serious issues. The conclusion section is missing, motivation is missing and the paper needs to be rewritten.

Author response: We sincerely thank you for the professional comment. We added a detailed motivation and conclusion for the proposed scheme.

Author action: In the revision, we updated the manuscript by meticulously proofreading and correcting these errors and mistakes, while also we added a detailed motivation (Page 2) and conclusion (Page 16) for the proposed scheme. Our modifications are as follows:

 

Comments of reviewers

Reviewers:1

Comments to the author

The authors seek to propose a broadcast signcryption scheme supporting equality test based on certificateless Cryptosystem for smart grid. I have the following suggestions that need to be answered.

emerging from their work they should address:

Reviewer#1, Conceren#1: 

In the abstract, the sentence needs to be revised. “However, the current scheme cannot ensure user privacy or efficient network bandwidth utilization.”

Author response: We are very grateful for this suggestion. According to your suggestion, we revised the sentence “However, the current scheme cannot ensure user privacy or efficient network bandwidth utilization.”

Author action: In the revision, we revised the sentence “However, the current scheme cannot ensure user privacy or efficient network bandwidth utilization.” to “However, most of the existing schemes require a large amount of network bandwidth resources and cannot ensure the receiver's anonymity.”

Reviewer#1, Conceren#2: 

I would suggest the authors to add a detailed motivation for the proposed scheme.

Author response: We sincerely thank you for the professional comment. We added a detailed motivation for the proposed scheme.

Author action: In the revision, we added a detailed motivation for the proposed scheme in motivation and contributions (Page 2). Our modifications are as follows:

Reviewer#1, Conceren#3: 

A detailed paragraph regarding the hardness efficiency details of the proposed scheme needs to be added above the contribution’s headings. The authors need to show why the proposed scheme is efficient and how.

Author response: We are extremely grateful for your comment. We added the hardness efficiency details of the proposed scheme to show why the proposed scheme is efficient and how in the introduction.

Author action: In the revision, we added the hardness efficiency details of the proposed scheme to show why the proposed scheme is efficient and how in Section 1 (Page 2). Our modifications are as follows:

Reviewer#1, Conceren#4: 

The entire literature review section is written in the present tense. The words like constructed, presented, and proposed need to be changed to propose, construct and present.

Author response: We are very grateful for this suggestion. According to your valuable suggestion, we have carefully revised our manuscript. We changed words like constructed, presented and proposed using the present tense.

Author action: We changed words like constructed, presented and proposed using the present tense (Page 2). Our modifications are as follows:

Reviewer#1, Conceren#5: 

The related work is limited; I would suggest the authors expand the related work. Besides, I would suggest the authors to conclude the related work. Currently, the authors did not conclude the related work; what did they learn after reviewing the literature?

Author response: Thank you very much for your invaluable suggestion. We expanded the related works and added the conclusion of related works in Section 2.

Author action: In the revision, we expanded the related works and added the conclusion of related works in Section 2 (Page 3). Our modifications are as follows:

Reviewer#1, Conceren#6: 

In the system model, the authors just define the terminology used in the scheme; however, the workflow of the proposed scheme is missing.

Author response: We are very grateful for this suggestion. According to your valuable suggestion, we have carefully revised our manuscript. We added the workflow of the proposed scheme to the system model.

Author action: In the revision, we added the workflow of the proposed scheme in the system model (Page 4). Our modifications are as follows:

Reviewer#1, Conceren#7: 

The scheme's definition needs to be named with the subsection, like System Definition.

Author response: We sincerely thank you for the professional suggestion and comment. We remove the scheme’s definition from the system model subsection and create a subsection for our scheme’ definition.

Author action: We remove the scheme’s definition from the system model subsection and create a subsection for our scheme’s definition (Page 5). Our modifications are as follows:

Reviewer#1, Conceren#8: 

I haven’t seen any details in the experimental analysis for the number of devices. How and from where did the authors add the details for the number of devices? How can the authors set the devices hypothetically? The authors did not conclude their research; a conclusion is necessary and must be added.

Author response: We sincerely thank you for the professional suggestion and comment. We simulate the experiment using bilinear pairing-based cryptography library under the Linux operating system. So the number of devices is manually set according to the existing scheme settings. In the experiment, the number of devices can also be used to indicate the number of smart meters in the smart grid. The number of devices on the smart meter can be dynamically adjusted to manage authorized devices more flexibly. Based on the reviewer’s suggestion, we modified the number of devices to the number of smart meters in Fig 3, Fig 4, Fig 5 and added the conclusion to the manuscript.

Author action: In the revision, we modified the number of devices to the number of smart meters in Fig 3, Fig 4, Fig 5 and added the conclusion of our research to the manuscript in Section 7 (Page 16). Our modifications are as follows:

Reviewer#1, Conceren#9: 

The article needs to be thoroughly proofread to remove all the grammatical and typos.

Author response: We apologize for the language problems in the original manuscript. According to the reviewer’s suggestions, we have corrected the errors and mistakes with due diligence and asked a native English speaker to edit and enhance the standard of English of our manuscript.

Author action: We updated the manuscript by meticulously proofreading and correcting these errors and mistakes. This revision is highlighted in blue in the text. 

Reviewers:2

Comments to the author

The technical aspects of the paper are fine. The authors, however, will need to make some minor adjustments before publishing it. The following are some of the concerns that the authors should address:

Reviewer#2, Conceren#1: 

The introduction is quite brief. The authors should add to it, highlighting the smart grid's vulnerabilities and the merits of using a certificateless signcryption scheme.

Author response: We are very grateful for this suggestion. According to your valuable suggestion, we have carefully revised our manuscript. We added the smart grid’s vulnerabilities and the merits of using a certificateless signcryption scheme in introduction.

Author action: In the revision, we added the smart grid’s vulnerabilities and the merits of using a certificateless signcryption scheme in the introduction (Page 2). Our modifications are as follows:

Reviewer#2, Conceren#2: 

Once more, the literature review section is extremely brief. Few relevant articles precisely on the same subject are ignored, for example (https://doi.org/10.3390/su131910891). The authors must contribute a few additional articles to this section.

Author response: We are very grateful for this suggestion. According to your valuable suggestion, we added the literature review in Section 2.

Author action: In the revision, we added the relevant articles precisely on the same subject in Section 2 (Page 3). Our modifications are as follows:

Reviewer#2, Conceren#3: 

I could not locate the section's conclusion. I have no idea why the conclusion is missing. Authors must provide justification.

Author response: We sincerely thank you for the professional comment. According to your suggestion, we added the conclusion to the manuscript.

Author action: In the revision, we added the conclusion in Section 7 (Page 16). Our modifications are as follows:

Reviewer#2, Conceren#4: 

The authors need to revise the article with correct usage of English, grammatical mistakes, and punctuation errors.

Author response: We apologize for the language problems in the original manuscript. According to the reviewer’s suggestions, we have corrected the errors and mistakes with due diligence and asked a native English speaker to edit and enhance the standard of English of our manuscript.

Author action: We updated the manuscript by meticulously proofreading and correcting these errors and mistakes. This revision is highlighted in blue in the text.

We are very grateful for your valuable comments. We carefully revised the manuscript following the Reviewers’ suggestions, tried our best to improve the manuscript, and made some changes in the manuscript. These changes will not influence the content and framework of the paper. We appreciate Editors/Reviewers’ warm work earnestly and hope that the correction will meet with approval. Once again, thank you very much for your comments and suggestions.

---

## [Decision Letter · Decision Letter 1]

13 Aug 2023

Certificateless broadcast signcryption scheme supporting equality test in smart grid

PONE-D-23-08826R1

Dear Dr. Dong,

We’re pleased to inform you that your manuscript has been judged scientifically suitable for publication and will be formally accepted for publication once it meets all outstanding technical requirements.

Kind regards,

Pandi Vijayakumar, Ph.D

Academic Editor

PLOS ONE

Additional Editor Comments (optional):

Reviewers' comments:

Reviewer's Responses to Questions

**Comments to the Author**

1. If the authors have adequately addressed your comments raised in a previous round of review and you feel that this manuscript is now acceptable for publication, you may indicate that here to bypass the “Comments to the Author” section, enter your conflict of interest statement in the “Confidential to Editor” section, and submit your "Accept" recommendation.

Reviewer #1: All comments have been addressed

2. Is the manuscript technically sound, and do the data support the conclusions?

Reviewer #1: Yes

3. Has the statistical analysis been performed appropriately and rigorously? 

Reviewer #1: Yes

4. Have the authors made all data underlying the findings in their manuscript fully available?

Reviewer #1: Yes

5. Is the manuscript presented in an intelligible fashion and written in standard English?

Reviewer #1: Yes

6. Review Comments to the Author

Reviewer #1: I don't have further comments. The authors addressed all my comments. I recommend the paper for a possible publication in Plos One.

7. PLOS authors have the option to publish the peer review history of their article (what does this mean?). If published, this will include your full peer review and any attached files.

Reviewer #1: No

---

## [Editor Report · Acceptance letter]

25 Aug 2023

PONE-D-23-08826R1 

Certificateless broadcast signcryption scheme supporting equality test in smart grid 

Dear Dr. Dong:

I'm pleased to inform you that your manuscript has been deemed suitable for publication in PLOS ONE. Congratulations! Your manuscript is now with our production department. 

Kind regards, 

on behalf of

Dr. Pandi Vijayakumar 

Academic Editor

PLOS ONE